**Data Availability Statement:** Data requests can be send to the Heidelberg University Hospital Ethics Committee (ethikkommission-I@med.uni-

# "Because at school, you can become somebody" – The perceived health and economic returns on secondary schooling in rural Burkina Faso

Luisa K. Werner[1,2], Jan Jabbarian[1], Moubassira Kagoné[1,3], Shannon McMahon[1,4], Julia Lemp[1], Aurélia Souares[1], Günther Fink[5,6‡], Jan-Walter De Neve[1‡]*

1 Heidelberg Institute of Global Health (HIGH), Medical Faculty and University Hospital, University of Heidelberg, Heidelberg, Germany, 2 Faculty of Medicine, Albert-Ludwigs-Universität Freiburg, Freiburg, Germany, 3 Health Research Centre of Nouna (Centre de Recherche en Santé de Nouna—CRSN), Ministry of Health, Nouna, Burkina Faso, 4 Bloomberg School of Public Health, Johns Hopkins University, Baltimore, United States, 5 Swiss Tropical and Public Health Institute, Basel, Switzerland, 6 University of Basel, Basel, Switzerland

‡ These authors are joint senior authors on this work.
* janwalter.deneve@uni-heidelberg.de

## Abstract

### Background

The perceived returns on schooling are critical in schooling decision-making but are not well understood. This study examines the perceived returns on secondary schooling in Burkina Faso, where secondary school completion is among the lowest globally (<10%).

### Methods

We conducted a two-staged qualitative study using semi-structured interviews ($N = 49$). In the first stage, we sampled students, dropouts, parents and teachers from a random sample of five schools ($n = 39$). In the second stage, we interviewed key informants knowledgeable of the school context using snowball sampling ($n = 10$). Systematic analysis was based on a grounded theory approach with a reading of transcripts, followed by coding of the narratives in NVivo 12.

### Results

Respondents nearly universally perceived health benefits to schooling. In particular, key health benefits included improved sexual and reproductive health outcomes, hygiene knowledge and practices, as well as better interactions with the formal health system. Common economic returns on schooling included improved employment opportunities and the provision of support to family members, in addition to generally attaining success and recognition. Indirect and long-term health returns, however, were infrequently mentioned by respondents.

heidelberg.de) by researchers who meet the criteria for access to confidential data. Interview transcripts cannot be made publicly available since they contain potentially sensitive information on e.g., out-of-school adolescents.

**Funding:** JWDN was supported by the Alexander von Humboldt Foundation, funded by Germany's Federal Ministry of Education and Research (https://www.humboldt-foundation.de/web/home.html); German Science Foundation (project nr: 405898232) (https://www.dfg.de/en/); and the Heidelberg University Excellence Initiative (https://www.uni-heidelberg.de/excellenceinitiative/). JWDN also acknowledges financial support by Deutsche Forschungsgemeinschaft within the funding programme Open Access Publishing, by the Baden-Württemberg Ministry of Science, Research and the Arts and by Ruprecht-Karls-Universität Heidelberg (https://www.ub.uni-heidelberg.de/Englisch/service/openaccess/publikationsfonds.html). The funders had no role in study design, data collection and analysis, decision to publish, or preparation of the manuscript.

**Competing interests:** The authors have declared that no competing interests exist.

## Conclusions

While respondents reported nearly universally short-term health benefits to schooling, responses with regard to economic as well as indirect and long-term health benefits were more ambiguous. Future intervention studies on the perceived returns on formal education are needed to inform policy and reach education and health targets in the region.

## Background

Secondary schooling has been linked with sustained benefits for health, economic and social outcomes [1], including reductions in mortality [2], and reductions in the risk of cardiovascular [3, 4] and sexually transmitted diseases [5, 6]. Investment in secondary schooling may not only yield large dividends for individuals themselves–but also for other household members [7, 8]. Given the strong links between secondary schooling and general well-being, as well as the progress made in primary schooling, policymakers have increasingly focused on promoting secondary school enrollment. The Sustainable Development Goals (SDGs), for instance, aim to achieve universal secondary education; whereas the recent *Let Girls Learn* and *Secondary Education Expansion for Development* (SEED) initiatives have focused on reducing gender inequality in secondary school attainment [9]. Despite major policy efforts, however, about one out of three children do not attend secondary schooling today, with about half of out-of-school children living in Sub-Saharan Africa [10].

In terms of the determinants of secondary schooling, the perceived returns on schooling have been suggested to be more important than measured returns on schooling [11]. Parents, for instance, may perceive smaller returns on education for girls than for boys, given the possibility of pregnancy during secondary school or gender discrimination in the labor market, leading them to underinvest in the school enrollment of their daughters [12]. In Madagascar, the provision of information on the benefits of schooling improved school attendance and test scores, and increased investment in children's human capital [13]. Similar results were found in experiments in the Dominican Republic [11], Kenya [14], Malawi [12], and in Serbia [15]. In the United Kingdom, a ten percentage points increase in the perceived returns on schooling among parents was associated with about 2.5 hours more time spent weekly with the child on average (such as helping out with homework, reading stories, and playing with their children), and an increase of about USD 15 monthly expenditure per child [16].

Understanding and 'adjusting' the perceived returns on schooling may be of particular importance in settings with limited resources [11, 13, 17], where social norms and attitudes may lead to differential investments in children's human capital [12], and where the perceived returns are comparatively low despite high real measured returns [18]. The perceived returns on schooling may also matter more during "critical periods" of development [19], when investments in human capital have disproportionally large effects on health and socio-economic outcomes across the life course [20]. During late adolescence, for instance, people develop new behavioral patterns and skills [21]; they make their own decisions for the first time; and many of these decisions have particular high path dependence, including pregnancy [22]. In this context, providing information on the health and economic benefits of schooling to school-going age children and their parents may yield particularly large returns.

### The present study

In this study, we examine the perceived returns on secondary schooling among nearly 50 informants (children and youth, parents, teachers, and key informants) in rural Burkina Faso,

where secondary school completion is among the lowest in the world (<10%) [23]. We con-
tribute to the literature on the perceived returns on schooling in three ways: i) we examine
*both* perceived health and economic returns on schooling, as opposed to either economic ben-
efits [16, 24, 25] or specific health domains (such as sexual and reproductive health [12]); ii)
we include the perspectives of parents, students, teachers, and *out-of-school* adolescents and
their parents–a typically hard-to-reach and vulnerable population–, as opposed to including
exclusively in-school adolescents and their parents [11, 25]; and, lastly, iii) we focus on the
period of *late adolescence*, a critical period for development with health and economic conse-
quences over the life course [20, 21]. To our knowledge, this study is among the first to com-
prehensively examine the hopes and expectations of secondary schooling, a critical element to
inform future interventions and reaching targets for education and health in the region.

## Materials and methods

### Study area

The study took place in the Health and Demographic Surveillance System (HDSS) of Nouna
in the Kossi province in the north-west of Burkina Faso (Boucle du Mouhoun region) (see **S1
File** with online supplementary materials for a map of the HDSS site). The HDSS has existed
since 1992 and currently covers a population of 105,000 habitants [26]. The Nouna HDSS
includes the semi-urban village of Nouna (29% of the population) as well as 58 villages (71% of
the population), with a total of 28 secondary schools. The mostly rural population of the study
area consists predominantly of subsistence farmers and cattle keepers. The Dioula language
serves as a lingua franca, permitting communication across ethnic groups. The Nouna HDSS
is operated by the Health Research Centre of Nouna (*Centre de Recherche en Santé de Nouna
—CRSN*), which provided the necessary infrastructure to conduct the current study. Addi-
tional details on the Nouna HDSS are provided elsewhere [27].

### Burkina Faso education system

The public education system in Burkina Faso, a landlocked country in western sub-Saharan
Africa, is based on the "6-4-3" system, including six years of primary education, four years of
junior secondary school (*Collège*), and three years of senior secondary school (*Lycée*). Burkina
Faso had the 7th lowest Education Index globally in 2013 (181st out of 187 countries) [28].
Gross lower secondary school enrollment ratio was 52% and upper secondary school enroll-
ment ratio was 17% in 2017. Mean years of schooling completed is one of the lowest in the
world, with a substantial gender gap (1.4 years of schooling completed among adults aged 25+;
1.0 among women and 1.9 among men in 2014) [23]. Access to school varies substantially by
geographical region, and is particularly low in rural areas [29]. The junior secondary school
completion rate, for instance, was 43% in the Centre region compared to 4.5% in the rural
Sahel region (see **S1 File** for additional information). A key reason for not enrolling in school
is that formal education is "not deemed necessary", suggesting that the perceived returns on
school may play an important role in schooling decision-making [30]. Other commonly
reported reasons for school absenteeism include financial barriers and distance to school.

### Study design

We conducted a two-staged qualitative study using semi-structured interviews. In the first
stage, we purposively sampled students, dropouts, parents and teachers from a random sample
of five secondary schools in the Nouna HDSS. In the second stage, we drew from a pool of
respondents as informed by guidance from local study team members. We complemented this

approach with snowball sampling and ultimately interviewed respondents including: school administrators, members of a students' parent association, home tutors, and health workers. The interviews across both phases asked about the perceived health and economic returns on secondary schooling. Topics included major health outcomes (such as sexual and reproductive health, mental health, violence, and injury); knowledge; norms, attitudes, and behaviors (such as alcohol and drug use); health services utilization; employment opportunities and salary; as well as the benefits of schooling among children for other household members, such as increased ability to support older family members (i.e. "upward" spillover effects of education) [7, 31]. The interviews also asked about hypothetical interventions which may reduce barriers to formal education, including changes in the perceptions of the benefits of attending secondary school. Prior to the interviews, we collected basic socio-demographic data of respondents in addition to structural facts about each selected school (e.g., number of students and teachers).

## Data collection

**Interviews.**   To help ensure consistency and completeness, we wrote semi-structured interview guides for each group of key respondents. Study instruments included sections on experiences with school, family environment, the relation between education and health, interactions with the formal health system, future plans and expectations, as well as potential spillover effects of children's education to family members and friends. The first-stage questionnaires were developed in English, translated to French, and then verified for comprehensibility by collaborators in Nouna, Burkina Faso. Second-stage questionnaires were designed by the study coordinator in the field based on first-stage questionnaires and informed by preliminary results from first-round interviews. Data collection in Nouna was implemented between April 14th and April 22nd 2018 by local interviewers from the CRSN with over 15 years of research experience. All data collectors received a two-day training session prior to field work, including training on sensitive points inherent to the study (e.g., how to interview adolescents who have engaged in a behavior that could be stigmatized, such as leaving school). Interviews were conducted in French, Dioula, or a mix of (local) languages with professional interpretation, according to interviewee's preferences and language proficiency. Interviews were audio-recorded for analysis and transcribed in French. Interviewees were informed of the purpose of the study, of our intention to take detailed notes of each interview, and of our process for handling interview data. All data collection activities took place at school or in the household, preferring a separate room to ensure confidentiality during the interview and discussions.

**Study sample.**   For the first stage, we drew a random sample of five secondary schools in the HDSS by drawing paper slips. Two sampled schools were in semi-urban Nouna and three in the rural surroundings. All sampled schools were at the junior secondary level by chance. We used maximum variation sampling within each school to capture large variability in the perceived benefits of schooling [32, 33]. Dropout adolescents, for instance, may have different perceptions of the benefits of schooling, vis-à-vis teachers or parents of in-school youth. From each selected school, a total of eight interviewees were selected: two enrolled students, two dropouts (dropped out of school between one month and one year prior to the interview), one parent of an enrolled student (independent from the chosen students), one parent of an out-of-school child, as well as two teachers. Inclusion and exclusion criteria are shown in **S1 File**. The sample size was guided by the work of Morse (2016) [34]: a total of about 50 respondents was considered sufficient to gain saturation as returns on schooling in Burkina Faso represent explicit, apparent information which does not focus on sensitive topics (e.g., adolescent pregnancy). The sample of parents was chosen to be smaller due to higher expected quality of

interviews with adults compared to those with children and youth. Respondents were randomly selected by data collectors from the most recent class lists available with the support of school staff as needed (e.g., for reaching dropouts). In the second stage of the study, additional interviews were conducted in the HDSS until saturation was reached [34, 35]. Five interviewees were recommended by participants from stage one using snowball sampling, and five interviews were identified by the research team as likely knowledgeable about the schooling and health context in Burkina Faso, with the help of local collaborators. One dropout could not be interviewed due to migration out of the HDSS and two transcripts were incomplete.

### Data analysis

Our analysis proceeded in two steps. The first step was content analysis informed by grounded theory [33, 36, 37]. We did not impose a framework but developed one in an inductive approach building on our coding results (**Fig 1**) [37]. Interview passages were coded to mutually exclusive topics (codes) arising from the interview data, using a codebook with descriptions for each topic to facilitate consistent use of codes by study team members throughout the study period. Major topics that arose from the data included health knowledge acquired at school (such as about general hygiene); interactions with the formal health system; language skills; employment; success and recognition; quality of life; as well as increased ability to provide support to family members and, in particular, parents in their old-age. Although codes were mutually exclusive, one interview passage could be applied to several different codes (a method called "simultaneous coding") [38]. The second step in our analysis was to group coded items by major categories based on commonalities and patterns. Major categories included health, economic factors, and key abilities and skills [39]. Lastly, we abstained from including frequencies of responses (a hierarchy of responses) since qualitative research is an iterative process–in contrast to quantitative research. This leads to a different depth between

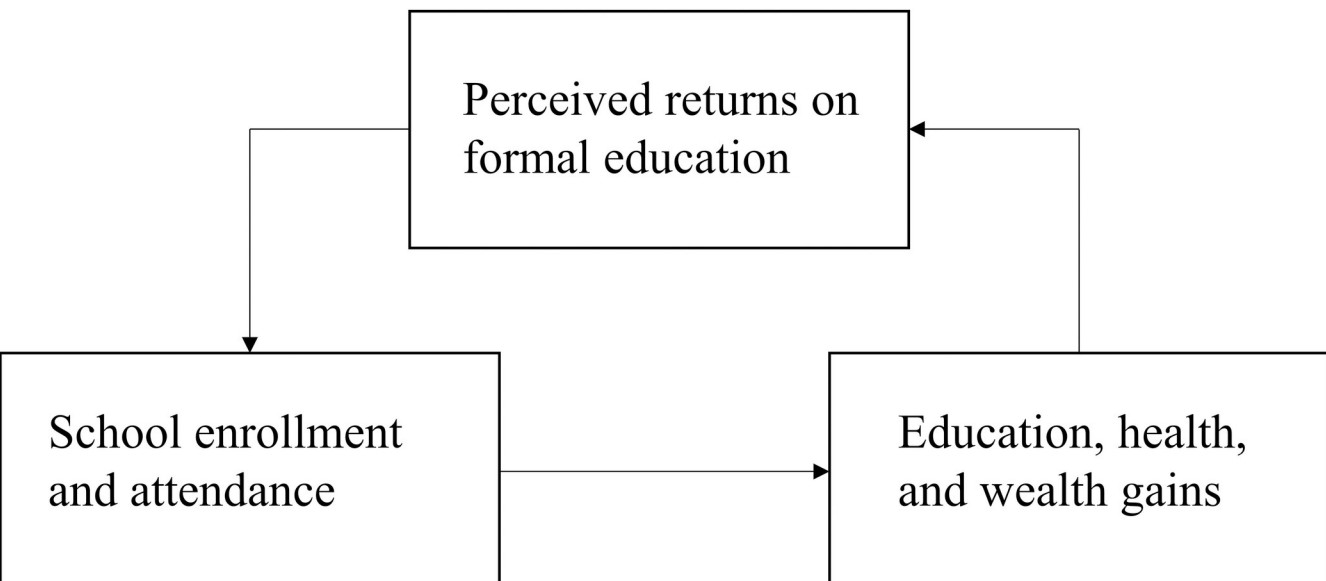

**Fig 1. Conceptual framework underpinning the study.** *Notes*: Authors' conceptual framework where the perceived benefits of secondary school may impact school enrollment and attendance and in turn the measured education (e.g., increased educational attainment), health (e.g., improved sexual and reproductive health outcomes), and household financial gains (e.g., increased employment opportunities, higher salary, and economic independence). The perceived benefits of formal education may also impact schooling during different periods across the life course (e.g., as child and parent) and for different stakeholders (e.g., children's household and family members).

the first and last interviews as one learns along the process. Using semi-structured interviews, not every participant was asked exactly the same questions, or was asked the questions with the same intensity or same probing. This would lead to a distortion of frequency results. All analyses were conducted using NVivo (QSR International Pty Ltd. Version 12, 2018).

### Ethical clearance

The study was approved by the *Comité Institutionnel d'Ethique du Centre de Recherche en Santé de Nouna* (N˚ 2018-03-/CIE-CRSN) and the Heidelberg University Hospital Ethics Committee (S-193/2018). Written informed consent was obtained from all adult respondents. Written parental consent was obtained from those younger than 18 years.

## Results

### Descriptive statistics

We interviewed a total of 49 participants, including children and youth (n = 19), parents (n = 10) and teachers (n = 10) across five schools in the Nouna HDSS. A number of other key informants were identified (n = 10), including school directors, the president of a students' parent association, home-based tutors, as well as healthcare workers. Average age was 26 years. Of the 49 respondents in the sample, 27% were female, 57% were Muslim, 39% were Christian, and 80% spoke French. Commonly spoken mother tongues included Bwamu (27%), Dafing (18%), Mooré (18%) and Dioula (10%). Overall, 17% of participants had no formal education. Average school fees per year were about USD 60, ranging from USD 13 to USD 130 per year depending on school level and school type (e.g., junior vs. senior secondary school). Nine out of 10 of the interviewed parents were employed in the agricultural sector, and most parents had no formal education whatsoever with only one parent having attended secondary school. The average number of siblings among students was 5 (range: 2–7). Further characteristics of the study participants are shown in **Table 1**. Selected characteristics of study schools are shown in **S1 File**. We report our results for the perceived returns on schooling below, separately for perceived health and economic benefits of schooling. Lastly, we examine heterogeneity in perceptions by gender; as well as by school enrollment status and generations (**S1 File**).

### Perceived health benefits to secondary schooling

Most respondents saw a direct connection between formal schooling and improved health outcomes. We provide an overview of perceived benefits and list specific examples in **Table 2**. Themes that were most frequently mentioned included sexual and reproductive health, general hygiene and health behavior, as well as the etiology, symptoms, and prevention of specific diseases. Sex education, in particular, was frequently mentioned by respondents, which was reported to be a taboo in family settings. Interviewees felt that learning about sexual and reproductive health topics at school was useful for family planning, the prevention of sexually transmitted infections, as well as for learning about the physiology of the body.

> *"There are others who do not know the date of their period, they do not know how many days it takes and at which day it ends. Through school you can get to know which day, or better said, how many days it takes, and at which day the next period starts, you can calculate and find out that day."* (23 years, female, enrolled student)

Schooling was also reported to play a critical role for water, sanitation and hygiene knowledge and practices (WASH). Respondents reported learning about general hygiene in school,

**Table 1. Selected characteristics of study respondents (*N* = 49).**

| Characteristics | Children and youth (n = 19) | | Parents (n = 10) | | Teachers (n = 10) | Other (n = 10) | Total n (%) |
|---|---|---|---|---|---|---|---|
| | *Dropout* | *Enrolled* | *Dropout* | *Enrolled* | | | |
| Age | | | | | | | |
| <15 | 0 | 2 | 0 | 0 | 0 | 0 | 2 (4) |
| 15–17 | 6 | 4 | 0 | 0 | 0 | 0 | 10 (20) |
| 18–20 | 2 | 0 | 0 | 0 | 0 | 1 | 3 (6) |
| 21–40 | 0 | 2 | 1 | 0 | 9 | 3 | 15 (31) |
| >40 | 0 | 0 | 3 | 3 | 0 | 5 | 11 (22) |
| Missing | 1 | 2 | 1 | 2 | 1 | 1 | 8 (16) |
| Gender | | | | | | | |
| Female | 4 | 5 | 0 | 0 | 3 | 1 | 13 (27) |
| Male | 5 | 5 | 5 | 5 | 7 | 9 | 36 (73) |
| Missing | 0 | 0 | 0 | 0 | 0 | 0 | 0 (0) |
| Religion | | | | | | | |
| Muslim | 6 | 4 | 3 | 4 | 4 | 7 | 28 (57) |
| Christian | 3 | 6 | 2 | 0 | 5 | 3 | 19 (39) |
| Animist | 0 | 0 | 0 | 1 | 1 | 0 | 2 (4) |
| Missing | 0 | 0 | 0 | 0 | 0 | 0 | 0 (0) |
| School level (years) | | | | | | | |
| 0 | 0 | 0 | 2 | 4 | 0 | 2 | 8 (16) |
| 1–6 | 0 | 0 | 2 | 0 | 0 | 1 | 3 (6) |
| 7–10 | 9 | 10 | 0 | 1 | 0 | 0 | 20 (41) |
| 11–13 | 0 | 0 | 0 | 0 | 0 | 1 | 1 (2) |
| >13 | 0 | 0 | 0 | 0 | 10 | 6 | 16 (33) |
| Missing | 0 | 0 | 1 | 0 | 0 | 0 | 1 (2) |
| Mother tongue | | | | | | | |
| Bwamu | 3 | 4 | 2 | 1 | 2 | 1 | 13 (27) |
| Dafing | 4 | 2 | 1 | 1 | 0 | 1 | 9 (18) |
| Mooré | 0 | 1 | 0 | 1 | 2 | 5 | 9 (18) |
| Dioula | 1 | 0 | 1 | 0 | 1 | 2 | 5 (10) |
| Other | 1 | 3 | 1 | 2 | 5 | 1 | 13 (27) |
| Missing | 0 | 0 | 0 | 0 | 0 | 0 | 0 (0) |
| French | | | | | | | |
| Yes | 7 | 10 | 3 | 0 | 10 | 9 | 39 (80) |
| No | 2 | 0 | 2 | 5 | 0 | 1 | 10 (20) |
| Missing | 0 | 0 | 0 | 0 | 0 | 0 | 0 (0) |

Table shows selected characteristics of study respondents. Data are number of individuals. The category 'Other' includes key informants who were interviewed during the second stage of data collection and include students, tutors, teachers, school directors, member of a students' parent association, as well as health workers. Structural characteristics on the secondary schools included in the study are available in **S1 File**.

including handwashing before eating, and refraining from drinking dirty water. Parents, teachers, and key informants from local health facilities noted that schooled adolescents appeared more sanitary than those without formal education. Knowledge acquired in secondary school was considered helpful to prevent specific diseases, such as tetanus and measles.

**Table 2. Perceived health and economic benefits of secondary schooling.**

| Benefit | | Examples provided by respondents | Related concerns and trade-offs |
|---|---|---|---|
| *A. Health* | *Behavior* | • Handwashing before eating<br>• Drinking clean water<br>• Clean physical appearance<br>• Avoidance of risky behavior<br>• Delayed age at first marriage<br>• Deliberation, self-control, respect, honesty | • Carrying out pregnancy<br>• Health barriers to school (e.g., poor vision)<br>• Food insecurity at school and in community<br>• Fatigue (e.g., household chores, travel to school)<br>• Security concerns while travelling to school<br>• Need to move far away for school (e.g., to stay with a tutor) |
| | *Knowledge* | • Etiology, symptoms, and prevention of diseases<br>• Sexual and reproductive health (e.g., family planning)<br>• Sharing knowledge with family and friends | |
| | *Health system* | • Understanding medical prescriptions<br>• Faster health care seeking<br>• Less reliance on traditional medicine<br>• Direct referral from school to health services | • Informal interpreters at health centers |
| *B. Economic* | *Employment* | • Literacy and language skills<br>• Financial independence<br>• Stable employment in e.g., government<br>• Development of the country | • Pecuniary gains in the short term<br>• Informal employment in family's undertaking<br>• Lack of infrastructure for higher studies<br>• Lack of desired career track |
| | *Success, recognition* | • "Becoming someone"<br>• Preparedness for the "difficulties in life"<br>• High position in government | • Traditional upbringing<br>• Fear of grade repetition and failure |
| | *Spillover effects* | • Caring for family and friends<br>• Knowledge to help support family business<br>• Protecting others against abuse by authorities | • Saving school fees<br>• Supporting family through short-term employment |

*"Generally, when you see a group of students at some spot among others [who are not students], you can easily distinguish the students by their cleanliness." (35 years, male, teacher)*

Skills acquired at school, such as reading and writing, were also perceived as helpful for facilitating interactions with the formal health system. Schooling provided basic scientific knowledge to better understand medical prescriptions. Conversely, health workers mentioned that patients *without* formal education were at risk of medication errors (e.g., improper dosage). Knowing how to read and write as well as speaking a common language (French) was perceived as facilitating communication with health workers. The ability to communicate across ethnicities and better understand health messaging, improved an individual's safety and health outcomes. Risky behaviors, such as driving fast, were thought to be reduced as a result of improved knowledge and literacy skills acquired at school. Although interviewees did not report schooling to cause a significant difference in health system access, schooled individuals were described as consulting health services earlier, and relying less on traditional medicine compared to those without formal education. Respondents also reported that school administrators referred students directly to primary health care centers when students are sick.

*". . . those who have not been educated . . ., if they get a motorcycle, they do not know that there is a maximum speed for the motorcycle. They only think you must go fast. But for someone who went to school . . . and then they show you the maximum speed, and then there are written instructions. . . those instructions help a lot for those who are literate." (26 years, female, nurse)*

In addition to direct health returns on schooling, respondents noted more general benefits, including noncognitive skills such as self-control. Respondents reported that school provided "good values" in life, including respect, politeness, and honesty. Parents of enrolled children also reported less "lingering around" as a result of spending time at school or doing homework, which reduced adolescents' exposure to risky environments. Teachers were more cautious, however, and noted that whether there is an effect of schooling on student behaviors and attitudes ultimately relied on the student's implementation of knowledge and skills acquired at school. Awareness of the negative health effects of smoking or alcohol consumption alone, for instance, may not be enough to reduce risky health behaviors outside of the classroom.

*". . . if he [my son] wasn't at school, it would be difficult for him to take on a good attitude . . . because he would always just react according to what comes to his mind." (55 years, male, parent)*

A small number of interviewees did not perceive health benefits to secondary schooling; or seemed to not have immediately understood what the interviewer was referring to with the study's questions, suggesting that they might not see a direct link between formal schooling and health outcomes. Long-term and rather indirect health benefits were generally not linked to school attendance (such as increased old-age survival and reduced catastrophic health expenditures as a result of better employment opportunities and higher wages).

## Perceived economic benefits to secondary schooling

In contrast to the perceived health returns on schooling, the perceptions of economic returns were somewhat more ambiguous. On the one hand, participants judged schooling to improve future job opportunities, and particularly stable employment with the government. The professions which enrolled students aspired to were typically government official, teacher, policeman, and nurse (most of which require at least some secondary schooling). Parents wanted their children to "become someone", attain a high position in the government, and become financially independent. Nearly all parents named as a main expectation and goal of sending children to school having their children take care of them in the future.

*"Because if I am going to be successful, I can be a government official for my family." (13 years, male, student)*

We identified some tension, however, between career aspirations and the observed struggles of finding employment after graduation. Some respondents, for instance, criticized that there may not be enough employment opportunities for secondary school and university graduates, thereby reducing the economic returns of investments in schooling. Some interviewees reported that adolescents considered schooling an investment with little or no economic return.

*". . . I give the example of a friend, he bluntly told me that since he works at the bus station and gains 5,000 francs [~8 USD] per day, well he thinks that it is more important to keep working at the station and take home 5,000 francs [~8 USD] each day rather than go sit down for fourteen years without getting anything and become a beggar, a complete burden for your parents." (20 years, male, student)*

Similarly, we identified tension between short- and long-run economic gains. Schooling came with the opportunity cost of short-run pecuniary gains, even though incomes are usually

low in late adolescence. Teachers indicated that parents' expectation of earning an income and supporting their family was a frequent cause of school absenteeism. Most dropouts were currently working in the household or family's business (e.g., agriculture, kiosque). Parents and adolescents, however, did not confirm these views but rather identified a general lack of resources and motivation among adolescents as key causes of school absenteeism.

Competencies acquired at school, including literacy and language skills, were also conceived to be helpful in the context of professional life. School provided an opportunity to acquire sufficient proficiency in French needed for future training, including tertiary education, formal employment (e.g., as a government official), as well as informal employment. A common language was deemed crucial, especially considering the many languages spoken in Burkina Faso (respondents in our sample reported more than four different mother tongues). As French proficiency at the end of primary school was considered rather poor by teachers, secondary school would play an important role in further consolidating students' language skills.

*". . . if someone went to school, even if he didn't get a job, that [French] is a skill, too . . . If there are upcoming projects, you can participate. Where you earn something to eat, that also is an advantage. No one can take that skill away." (56 years, male, parent)*

Apart from purely economic returns, interviewees mentioned more general benefits of schooling such as increased success, recognition, and a "better life". Schooling was suggested to open up opportunities and allow increased participation in planning a future. Moreover, in addition to individual benefits, education was recognized as a base for economic development at the aggregate level. Adults noted the potential of schooling to develop the country by "educating the nation's sons" and spurring development of the family, village, and country.

*"Because at school, you can become somebody in the future, you can become somebody respectful, you can have a good future." (17 years, male, student)*

Respondents who dropped out of school were generally unhappy about the decision to leave school, and none of them expressed advantages of dropping out. They explained their decision rather in the context of specific barriers to school (e.g., lack of financial means and transport to school, poor performance and grade repetition, pregnancy, marriage). Most dropouts wanted to return to school of which one respondent wanted to attend a hairdresser school in the future.

### Heterogeneity in perceptions by gender

When comparing across genders, both female and male adolescents aspired equally to schooling. A key benefit shared by both was that school opened up the possibility of employment and future career opportunities. Nevertheless, the type of desired employment differed (e.g., females preferred to become a nurse vs. a policeman). Female adolescents also more often brought up barriers to secondary schooling compared to their male counterparts, such as the lack of (unisex) toilets and security concerns. Similarly, pregnancy and marriage were topics which came up exclusively among female adolescents and were reasons for dropping out of school. Females reported going to school while pregnant or with their own children. All female dropouts, however, aspired going back to school, further suggesting high perceived returns on school attendance regardless of several additional concerns. Results for heterogeneity by other socio-demographic characteristics are presented in **S1 File**.

## Discussion

Using data from nearly 50 in-depth interviews with students, dropouts, parents, teachers, school administrators, as well as other key informants, we examined the perceived health and economic returns on secondary schooling in rural Burkina Faso. Average educational attainment in the country is among the lowest in the world (about one year of schooling completed) [23]. Nevertheless, respondents reported a wide array of health and economic benefits of schooling. Commonly mentioned health benefits included improved sexual and reproductive health outcomes, hygiene knowledge and practices, as well as improved interactions with the formal health system. Long-term and indirect health benefits, such as increased old-age survival, however, were infrequently mentioned. One reason for this finding could be the abstract nature of some of these concepts and delayed benefits [40]. In terms of economic benefits, respondents generally stated improved employment opportunities, success, and recognition. Public employment, in particular, was highly desired by respondents, offering stable work and status within community. The notion of sharing resources with older family members was also predominant and seemed to be a major motivation for sending children to school. Indeed, human capital gains acquired through schooling are likely to be shared with the older generations in settings where social security services in old-age are limited [31]. Changes in the perceptions of the returns on schooling are also likely to inform schooling investment decisions by students and parents [11, 16]. In our study, a hypothetical intervention–such as increasing awareness around the benefits of attending formal schooling–was perceived as a helpful measure by respondents to further reduce school absenteeism. These findings further suggest that the perceived benefits of schooling may play an important role in school decision-making.

While respondents reported nearly universally health benefits to schooling, responses regarding economic benefits were more ambiguous. Adolescents and youth in secondary school expressed fears of not securing their desired employment after graduation. Respondents expressed concerns of finding employment that would match their (secondary school level) qualifications. This may be particularly the case in rural areas–about 70% of the Burkina Faso's total population–for whom the majority of household income comes from agricultural activities [41]. Similarly, we identified tension between short- and long-run economic gains [42]. Respondents reported that adolescents dropped out of school to start work sooner (such as agricultural activities at home), rather than completing secondary school, thereby forsaking the 'option' to improve career prospects in the future [43]. Students also travelled long distances to secondary school or sought housing elsewhere (e.g., to live with a tutor), leading to high indirect costs of school attendance, which further impacted the perceived economic returns. These findings are consistent with studies from other low-income contexts [44].

This study has a number of implications. First, a key question is to what extent the perceptions of the benefits of schooling influence schooling investment decisions. Students were generally more aware of the array of schooling benefits compared to school dropouts (**S1 File**), suggesting that students might perceive higher returns than dropouts, leading them to invest more in schooling. However, with the current study design, we cannot ascertain whether this is a reason for them to stay at school or rather a consequence of schooling (or both). The perceived benefits of schooling are likely dynamic and the result of multiple factors. Schooling decisions are also made in the context of a complex social environment [45]. Future mixed-methods and (quasi-)experimental studies must assess whether efforts to raise awareness of the health and economic benefits of schooling could be a feasible approach to increasing school enrollment and attendance [46, 47]. Second, the perceived benefits of schooling also differed by gender and generations, further suggesting the need for tailored interventions. Third, it is not clear to what extent the perceived benefits of schooling are accurate. The perceived returns,

for instance, may reflect 'optimistic bias', because of unrealistic optimism about future life events [48]. Future research could compare the perceived returns on schooling with the measured returns on schooling, using longitudinal data routinely collected by the Nouna HDSS.

## Strengths and limitations

A key strength of the study is its broad topical focus, including both perceived health and economic benefits, drawing upon the perspectives of a wide range of stakeholders. Prior studies in developing countries have focused on either students [11] or their parents [17], even though a large number of children and youth are out of school in developing settings. Previous studies therefore miss the perceptions of a particularly vulnerable population. Nevertheless, our study has several limitations. First, the sample we drew was recruited in and around 5 out of 28 secondary schools in the Nouna HDSS (comprising 58 villages and the town of Nouna), possibly missing perceptions from individuals living in underserved villages. Second, student respondents were sometimes chosen by school staff, who might have a preference of choosing certain students. Third, all parents interviewed were male, possibly due to norms and attitudes which considered the father as head of the family, thereby omitting maternal perceptions [49]. Fourth, the interview setting could not always assure complete intimacy given a lack of infrastructure in this low-resource setting. Similarly, respondents, in particular adolescents who dropped out of school, may have felt intimidated by interviewers who were (by definition) relatively highly educated. Fifth, translation from the local language to French may have led to loss of information, even though local transcribers were fluent in both languages.

## Conclusions

The perceived returns on schooling are critical in schooling decision-making and human capital investment but are not well understood. We examined the perceived returns on secondary schooling in Burkina Faso, where access to education is among the lowest globally. Respondents nearly universally perceived health benefits to schooling. In particular, key health benefits included improved sexual and reproductive health outcomes, hygiene knowledge and practices, as well as better interactions with the formal health system. Most common economic returns included improved employment opportunities, the provision of support to older family members, as well as attaining success and recognition. Indirect and long-term health returns, however, were infrequently mentioned. Intervention studies on the perceived returns on schooling are needed to inform policy and reach education and health targets in the region.

## Supporting information

**S1 File. Online supplementary materials.**
(DOCX)

## Acknowledgments

We thank study participants for their time.

## Author Contributions

**Conceptualization:** Luisa K. Werner, Jan Jabbarian, Moubassira Kagoné, Aurélia Souares, Günther Fink, Jan-Walter De Neve.

**Data curation:** Luisa K. Werner, Jan Jabbarian, Moubassira Kagoné, Günther Fink, Jan-Walter De Neve.

**Formal analysis:** Luisa K. Werner, Jan Jabbarian, Shannon McMahon, Julia Lemp, Aurélia Souares, Günther Fink, Jan-Walter De Neve.

**Funding acquisition:** Jan-Walter De Neve.

**Investigation:** Luisa K. Werner, Jan Jabbarian, Moubassira Kagoné, Günther Fink, Jan-Walter De Neve.

**Methodology:** Luisa K. Werner, Jan Jabbarian, Moubassira Kagoné, Shannon McMahon, Julia Lemp, Aurélia Souares, Günther Fink, Jan-Walter De Neve.

**Project administration:** Luisa K. Werner, Jan Jabbarian, Moubassira Kagoné, Günther Fink, Jan-Walter De Neve.

**Resources:** Jan Jabbarian, Jan-Walter De Neve.

**Software:** Luisa K. Werner, Jan Jabbarian, Jan-Walter De Neve.

**Supervision:** Moubassira Kagoné, Günther Fink, Jan-Walter De Neve.

**Validation:** Luisa K. Werner, Jan Jabbarian, Günther Fink, Jan-Walter De Neve.

**Visualization:** Luisa K. Werner, Jan Jabbarian, Günther Fink, Jan-Walter De Neve.

**Writing – original draft:** Luisa K. Werner, Günther Fink, Jan-Walter De Neve.

**Writing – review & editing:** Luisa K. Werner, Jan Jabbarian, Moubassira Kagoné, Shannon McMahon, Julia Lemp, Aurélia Souares, Günther Fink, Jan-Walter De Neve.

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
