## [Decision Letter · Decision Letter 0]

30 Aug 2019

PONE-D-19-16158

“Because at school, you can become somebody” – The perceived health and economic returns to secondary schooling in rural Burkina Faso

PLOS ONE

Dear Dr De Neve,

Thank you for submitting your manuscript to PLOS ONE. After careful consideration, we feel that it has merit but does not fully meet PLOS ONE’s publication criteria as it currently stands. Therefore, we invite you to submit a revised version of the manuscript that addresses the points raised during the review process.

We would appreciate receiving your revised manuscript by Oct 14 2019 11:59PM. To enhance the reproducibility of your results, we recommend that if applicable you deposit your laboratory protocols in protocols.io, where a protocol can be assigned its own identifier (DOI) such that it can be cited independently in the future. For instructions see: http://journals.plos.org/plosone/s/submission-guidelines#loc-laboratory-protocols

We look forward to receiving your revised manuscript.

Kind regards,

Lindsay Stark

Academic Editor

PLOS ONE

Journal Requirements:

Reviewers' comments:

Reviewer's Responses to Questions

**Comments to the Author**

1. Is the manuscript technically sound, and do the data support the conclusions?

Reviewer #1: Yes

Reviewer #2: Yes

2. Has the statistical analysis been performed appropriately and rigorously? 

Reviewer #1: N/A

Reviewer #2: N/A

3. Have the authors made all data underlying the findings in their manuscript fully available?

Reviewer #1: No

Reviewer #2: Yes

4. Is the manuscript presented in an intelligible fashion and written in standard English?

Reviewer #1: Yes

Reviewer #2: Yes

5. Review Comments to the Author

Reviewer #1: This paper provides an analysis of perceived benefits of secondary schooling in Burkina Faso. Among the strengths of this paper are its focus on secondary schooling; its inclusion of data from children who are in and also out of school, parents and diverse key informants; and its use of narratives to bring forward the voices and perspectives of local people. The paper also notes appropriately its limits regarding the small number of participants (e.g., only 7 school dropouts) and possible sampling biases.

The paper would be improved by more systematic attention to and analysis of gender dimensions. It would help to bring the findings related to gender out of the supplementary material and into the main body of the paper. Similarly, in developing recommendations, it would be useful to consider the likely need to develop different kinds of messages for women and men, respectively. Also, it would help to have a bit more analysis of the possible role of perceived benefits (and narratives surrounding them) in the causal nexus affecting decisions to go to school, stay in school, send children to school, etc. At present, the statements that perceived benefits may be causal or non-causal seem too binary and out of touch with what is likely a multi-causal system that extends over time and in which perceived benefits could have an influence at different points (e.g., decisions to send children to school, decisions to stay in school, decisions to work harder, etc.) and for different stakeholders (e.g., siblings, extended families, school age children, etc.) in a system of social decision making. It could also be useful to note further how, apart from the possible influence of perceived benefits, the perceived benefits may be inaccurate and reflect optimistic biases, among others.

Minor language improvements include:

p. 19, line 9: ‘intimated’ should be ‘intimidated’

Abstract and after: ‘returns to schooling’ should be ‘returns on schooling’ since the former could refer to going back to school rather than the benefits of school participation.

I hope these comments are useful in making revisions.

Reviewer #2: The topic for the paper is timely and of interest to diverse research and practice areas. I do however believe the authors might have included a more robust literature review including what reasons for school drop out in primary schooling are understood and how this may/not inform this research on secondary schooling.

I suggest more discussion around the sampling and particular why such a small number of parents included.

I suggest clarifying frequency of responses in the results section to get a better idea of how many times certain answers were given or at least to see which answers were more frequent as and perhaps rank answers in frequency to see which responses were most given vs. 1 or 2 people

The authors may benefit from expanding their discussion of grounded theory, why they chose this line of inquiry and whether such an approach should be replicated in future studies. What other methods might be used in future studies and why? Given that this hard to reach population is understudied please the authors may elaborate on what future studies might work best

The limitations section as it provides a well thought out commentary of its limits. In addition, the policy and practice implications are well considered.

6. PLOS authors have the option to publish the peer review history of their article (what does this mean?). If published, this will include your full peer review and any attached files.

Reviewer #1: No

Reviewer #2: No

---

## [Author Response · Author response to Decision Letter 0]

18 Sep 2019

Reviewers' comments:

Reviewer #1: 

This paper provides an analysis of perceived benefits of secondary schooling in Burkina Faso. Among the strengths of this paper are its focus on secondary schooling; its inclusion of data from children who are in and also out of school, parents and diverse key informants; and its use of narratives to bring forward the voices and perspectives of local people. The paper also notes appropriately its limits regarding the small number of participants (e.g., only 7 school dropouts) and possible sampling biases.

We thank the reviewer for these comments.

The paper would be improved by more systematic attention to and analysis of gender dimensions. It would help to bring the findings related to gender out of the supplementary material and into the main body of the paper. 

We agree that the gender dimensions are an important component and have now discussed these in more detail in the Results and Discussion section of the paper. As recommended by the reviewer, we have moved findings related to gender out of the supplementary material and into the main text.

“We examine heterogeneity in perceptions by gender; as well as by school enrollment status and generations (S2 Text).” (p.10, revised manuscript)

“Heterogeneity in Perceptions by Gender

When comparing across genders, both female and male adolescents aspired equally to schooling, enabling them to gain employment. Nevertheless, the type of desired employment differed (e.g., females preferred to become a nurse vs. policeman). Female adolescents more often brought up barriers to secondary schooling compared to their male counterparts, such as the lack of (unisex) toilets, and security concerns. Similarly, marriage and pregnancy were also topics which came up exclusively among female adolescents and were reasons for dropping out of school. Females reported going to school while pregnant and/or with their children. All female dropouts, however, aspired going back to school, suggesting high perceived returns of secondary school attendance, despite these several additional concerns.” (p. 17, revised manuscript)

Similarly, in developing recommendations, it would be useful to consider the likely need to develop different kinds of messages for women and men, respectively.

We agree with the reviewer and have now further considered this in the Discussion section of the paper. We have also moved relevant material from the supplementary materials to the main text.

“Future (quasi-)experimental studies must assess whether efforts to raise awareness of the health and economic benefits of schooling could be a feasible approach to increasing school enrollment and attendance [45, 46]. The perceived benefits of schooling also differed by gender and generations, further suggesting the need for tailored interventions.” (p. 19, revised manuscript)

Also, it would help to have a bit more analysis of the possible role of perceived benefits (and narratives surrounding them) in the causal nexus affecting decisions to go to school, stay in school, send children to school, etc. At present, the statements that perceived benefits may be causal or non-causal seem too binary and out of touch with what is likely a multi-causal system that extends over time and in which perceived benefits could have an influence at different points (e.g., decisions to send children to school, decisions to stay in school, decisions to work harder, etc.) and for different stakeholders (e.g., siblings, extended families, school age children, etc.) in a system of social decision making. It could also be useful to note further how, apart from the possible influence of perceived benefits, the perceived benefits may be inaccurate and reflect optimistic biases, among others.

We thank the reviewer for these excellent suggestions. We have now further clarified these points in the Methods and Discussions sections of the paper, where we write:

“A key question is to what extent the perceptions of the benefits of schooling influence schooling investment decisions. In our analysis, students were generally more aware of the array of schooling benefits compared to school dropouts (S2 Text), suggesting that students might perceive higher returns than dropouts, leading them to invest more in schooling. However, with the current study design, we cannot ascertain whether this is a reason for them to stay at school or rather a consequence of schooling (or both). The perceived benefits of schooling are likely dynamic and the result of multiple factors. Schooling decisions are also made in the context of a complex social environment [44]. Future (quasi-)experimental studies must assess whether efforts to raise awareness of the health and economic benefits of schooling could be a feasible approach to increasing school enrollment and attendance [45, 46]. The perceived benefits of schooling also differed by gender and generations, further suggesting the need for tailored interventions.” (p. 19, revised manuscript)

“It is not clear to what extent the perceived benefits of schooling are accurate. The perceived returns, for instance, may reflect “optimistic bias”, because of unrealistic optimism about future life events [43]. Future research could compare the perceived returns with the measured returns on schooling, using longitudinal data routinely collected by the HDSS.” (p. 19, revised manuscript)

“The perceived benefits may impact schooling during different periods across the life course (e.g., as adolescent or parent), and for different stakeholders (e.g., siblings, extended families).” (Figure 1, revised manuscript)

References:

Sanfey A.G. (2007). Social decision-making: insights from game theory and neuroscience. Science. 26;318(5850):598-602

Weinstein, N. D. (1980). Unrealistic optimism about future life events. Journal of Personality and Social Psychology, 39(5), 806-820.

Minor language improvements include: p. 19, line 9: ‘intimated’ should be ‘intimidated’

We thank the reviewer for bringing this to our attention and have now corrected this sentence:

“Similarly, respondents, in particular adolescents who dropped out of school, may have felt intimidated by interviewers who were (by definition) relatively highly educated.” (p.20, revised manuscript)

Abstract and after: ‘returns to schooling’ should be ‘returns on schooling’ since the former could refer to going back to school rather than the benefits of school participation.

We have now edited this throughout the entire paper – e.g., see the following paragraphs:

“Because at school, you can become somebody” – The perceived health and economic returns on secondary schooling in rural Burkina Faso” (p.1, revised manuscript)

“In this study, we examine the perceived returns on secondary schooling among nearly 50 informants (school-age adolescents, parents, teachers, and key informants) in rural Burkina Faso, where secondary school completion is among the lowest in the world (<10%) [22].” (p.4, revised manuscript)

“The perceived returns on schooling are critical in schooling decision making and human capital investment but are not well understood.” (p.20, revised manuscript)

I hope these comments are useful in making revisions.

We thank the reviewer for these comments.

Reviewer #2: 

The topic for the paper is timely and of interest to diverse research and practice areas. 

We thank the reviewer for these comments.

I do however believe the authors might have included a more robust literature review including what reasons for school dropout in primary schooling are understood and how this may/not inform this research on secondary schooling.

We thank the reviewer for this excellent suggestion. We have now provided additional discussion on reasons for the lack of access to formal schooling and school dropout, where we write: 

“A key reason for not enrolling in school is that formal schooling is “not deemed necessary”, suggesting that the perceived returns on attending formal schooling may play an important role in schooling decision-making [29]. Other commonly reported reasons for school absenteeism include financial barriers and distance to school.” (p.5, revised manuscript).

Reference:

National Institute of Statistics and Demography (2010), Enquete Integrale sur les Conditions de Vie des Menages [Integrated Household Living Conditions Survey], Burkina Faso, 2010.

I suggest more discussion around the sampling and particular why such a small number of parents included.

We have now further elaborated on our sampling approach in the Methods section (Morse 2016):

“The sample size was guided by the work of Morse (2016) [32]: 50 was considered as a number sufficient to gain saturation as returns on schooling in Burkina Faso represent explicit, apparent information which does not focus on sensitive topics (e.g., adolescent pregnancy). Furthermore, the sample of parents was chosen to be smaller due to higher expected quality of interviews with adults compared to those with adolescents.” (p.8, revised manuscript)

Reference:

Morse J.M. (2016). Determining Sample Size. Qualitative Health Research;10(1):3-5

I suggest clarifying frequency of responses in the results section to get a better idea of how many times certain answers were given or at least to see which answers were more frequent as and perhaps rank answers in frequency to see which responses were most given vs. 1 or 2 people.

We abstained from including frequencies of responses for the following reason. Qualitative research is an iterative process – in contrast to quantitative research – which questions rather than measures. This leads to a different depth between the first and last interviews as one learns along the process. Using semi-structured interviews, not every participant was asked exactly the same questions, or was asked the questions with the same intensity or same probing. This would lead to a distortion of frequency results. We would be happy, however, to make further changes at the behest of the editor.

Reference:

Auerbach, C. F., Silverstein, L. B. (2003). Qualitative data: An introduction to coding and analysis. New York, NY, US: New York University Press.

The authors may benefit from expanding their discussion of grounded theory, why they chose this line of inquiry and whether such an approach should be replicated in future studies. What other methods might be used in future studies and why? Given that this hard-to-reach population is understudied please the authors may elaborate on what future studies might work best.

We thank the reviewer for these excellent suggestions. We have now provided more discussion on our analytical approach. Additionally, we have now moved the theoretical framework (Figure 1), which underpins our study, out of the supplementary material and inserted it into the main text to further clarify our approach and findings. As recommended by the reviewer, we have also further elaborated on the implications of our findings for research, including possible future study designs.

“Our analysis proceeded in two steps. The first step was content analysis informed by grounded theory [32, 35, 36]. We did not impose a framework but developed one in an inductive approach building on our coding results (Figure 1) [36]. Interview passages were coded to mutually exclusive topics (codes) arising from the interview data, using a codebook with descriptions for each topic to facilitate consistent use of codes by study team members throughout the study period. Major topics that arose from the data included health knowledge acquired at school (such as about general hygiene); interactions with the formal health system; language skills; employment; success and recognition; quality of life; as well as increased ability to provide support to family members and, in particular, parents in their old-age. The second step in our analysis was to group coded items by major categories based on commonalities and patterns. Major categories included health, economic factors, and key abilities and skills [37].” (p.8, revised manuscript)

“It is not clear to what extent the perceived benefits of schooling are accurate. The perceived returns, for instance, may reflect “optimistic bias”, because of unrealistic optimism about future life events [43]. Future research could compare the perceived returns with the measured returns on schooling, using longitudinal data routinely collected by the HDSS.” (p.19, revised manuscript)

“A key question is to what extent the perceptions of the benefits of schooling influence schooling investment decisions. In our analysis, students were generally more aware of the array of schooling benefits compared to school dropouts (S2 Text), suggesting that students might perceive higher returns than dropouts, leading them to invest more in schooling. However, with the current study design, we cannot ascertain whether this is a reason for them to stay at school or rather a consequence of schooling (or both). The perceived benefits of schooling are likely dynamic and the result of multiple factors. Schooling decisions are also made in the context of a complex social environment [44]. Future (quasi-)experimental studies must assess whether efforts to raise awareness of the health and economic benefits of schooling could be a feasible approach to increasing school enrollment and attendance [45, 46]. The perceived benefits of schooling also differed by gender and generations, further suggesting the need for tailored interventions.” (p.19, revised manuscript)

“Figure 1. Conceptual framework underpinning the study.” (p.9, revised manuscript)

References:

Charmaz K. (2006). Constructing grounded theory. London; Thousand Oaks, Calif.: Sage Publications. xiii, 208 p.p.

Snilstveit B, Stevenson J, Phillips D, Vojtkova M, Gallagher E, Schmidt T, et al. (2015). Interventions for improving learning outcomes and access to education in low- and middleincome countries: a systematic review, 3ie Systematic Review 24. London.

The limitations section as it provides a well thought out commentary of its limits. In addition, the policy and practice implications are well considered.

We thank the reviewer for these comments.

---

## [Decision Letter · Decision Letter 1]

31 Oct 2019

PONE-D-19-16158R1

“Because at school, you can become somebody” – The perceived health and economic returns on secondary schooling in rural Burkina Faso

PLOS ONE

Dear Dr De Neve,

Thank you for submitting your manuscript to PLOS ONE. After careful consideration, we feel that it has merit but does not fully meet PLOS ONE’s publication criteria as it currently stands. Therefore, we invite you to submit a revised version of the manuscript that addresses the points raised during the review process.

We would appreciate receiving your revised manuscript by Dec 15 2019 11:59PM. To enhance the reproducibility of your results, we recommend that if applicable you deposit your laboratory protocols in protocols.io, where a protocol can be assigned its own identifier (DOI) such that it can be cited independently in the future. For instructions see: http://journals.plos.org/plosone/s/submission-guidelines#loc-laboratory-protocols

We look forward to receiving your revised manuscript.

Kind regards,

Lindsay Stark

Academic Editor

PLOS ONE

Reviewers' comments:

Reviewer's Responses to Questions

**Comments to the Author**

1. If the authors have adequately addressed your comments raised in a previous round of review and you feel that this manuscript is now acceptable for publication, you may indicate that here to bypass the “Comments to the Author” section, enter your conflict of interest statement in the “Confidential to Editor” section, and submit your "Accept" recommendation.

Reviewer #1: All comments have been addressed

Reviewer #2: All comments have been addressed

2. Is the manuscript technically sound, and do the data support the conclusions?

Reviewer #1: Yes

Reviewer #2: Partly

3. Has the statistical analysis been performed appropriately and rigorously? 

Reviewer #1: N/A

Reviewer #2: N/A

4. Have the authors made all data underlying the findings in their manuscript fully available?

Reviewer #1: Yes

Reviewer #2: (No Response)

5. Is the manuscript presented in an intelligible fashion and written in standard English?

Reviewer #1: Yes

Reviewer #2: Yes

6. Review Comments to the Author

Reviewer #1: (No Response)

Reviewer #2: Question on Table 1. Characteristics of study respondents, under Age of School-age adolescent Dropouts, there seems to be one subject not accounted for? the total of dropouts I believe was 9 however under ages I only count 8? There are 6 under ages 15-17, 1 ages 18-20 and 1 Missing however all other totals in that column equal 9

Also, again was there any analysis in Table 2 about which answers were most common (hierarchy of responses?) as I believe categories were exclusive?

7. PLOS authors have the option to publish the peer review history of their article (what does this mean?). If published, this will include your full peer review and any attached files.

Reviewer #1: No

Reviewer #2: No

---

## [Author Response · Author response to Decision Letter 1]

6 Nov 2019

Reviewer #2: 

Question on Table 1. Characteristics of study respondents, under Age of School-age adolescent Dropouts, there seems to be one subject not accounted for? the total of dropouts I believe was 9 however under ages I only count 8? There are 6 under ages 15-17, 1 ages 18-20 and 1 Missing however all other totals in that column equal 9

We thank the reviewer for bringing this to our attention. The age of one dropout ages 18-20 was not accounted for and we have now corrected this in Table 1 (so that the total in the corresponding column equals 9):

“Table 1. Selected characteristics of study respondents (N = 49)”

Was there any analysis in Table 2 about which answers were most common (hierarchy of responses?) as I believe categories were exclusive?

We abstained from including frequencies of responses in Table 2 as qualitative research is an iterative process – in contrast to quantitative research – which questions rather than measures. We have now further clarified our analysis in Table 2 in the Methods section of the paper, where we write:

“Lastly, we abstained from including frequencies of responses (a hierarchy of responses) since qualitative research is an iterative process – in contrast to quantitative research. This leads to a different depth between the first and last interviews as one learns along the process. Using semi-structured interviews, not every participant was asked exactly the same questions, or was asked the questions with the same intensity or same probing. This would lead to a distortion of frequency results.” (p.9, revised manuscript)

References:

Auerbach, C. F., Silverstein, L. B. (2003). Qualitative data: An introduction to coding and analysis. New York, NY, US: New York University Press.

Saldaña, J. (2009). The coding manual for qualitative researchers. Thousand Oaks, CA, US: Sage Publications Ltd.

---

## [Editor Report · Decision Letter 2]

10 Dec 2019

“Because at school, you can become somebody” – The perceived health and economic returns on secondary schooling in rural Burkina Faso

PONE-D-19-16158R2

Dear Dr. De Neve,

We are pleased to inform you that your manuscript has been judged scientifically suitable for publication and will be formally accepted for publication once it complies with all outstanding technical requirements.

With kind regards,

Lindsay Stark

Academic Editor

PLOS ONE
---

## [Editor Report · Acceptance letter]

18 Dec 2019

PONE-D-19-16158R2 

“Because at school, you can become somebody” – The perceived health and economic returns on secondary schooling in rural Burkina Faso 

Dear Dr. De Neve:

I am pleased to inform you that your manuscript has been deemed suitable for publication in PLOS ONE. Congratulations! Your manuscript is now with our production department. 

With kind regards,

on behalf of

Dr. Lindsay Stark 

Academic Editor

PLOS ONE